# A Kinetic Stem Cell Counting Analysis of the Specific Effects of Cell Culture Medium Growth Factors on Adipose-Derived Mesenchymal Stem Cells

**DOI:** 10.3390/life13030614

**Published:** 2023-02-23

**Authors:** James L. Sherley

**Affiliations:** Asymmetrex^®^, LLC, Boston, MA 02130, USA; jsherley@asymmetrex.com; Tel.: +1-617-990-6819

**Keywords:** tissue stem cell, kinetic stem cell counting, mesenchymal stem cell, adipose, fetal bovine serum, human sera, human platelet lysate, asymmetric self-renewal, stem cell expansion, stem cell fraction

## Abstract

A recently described kinetic stem cell (KSC) counting method was used to investigate the stem-cell-specific effects of commercial growth factor supplements used for expanding stem cells in adipose-tissue-derived mesenchymal cell preparations. The supplements were a proprietary growth factor product, a source of fetal bovine serum, two sources of pooled human sera, and two sources of human platelet lysate. KSC counting analyses were performed to monitor effects on the fraction and viability of stem cells in serial cultures with their respective supplements. Serial cultures supplemented with the proprietary growth factor product or fetal bovine serum showed a similar high degree of maintenance of stem cell fraction with passage. In contrast, cultures supplemented with human sera or human platelet lysate showed rapid declines in stem cell fraction. KSC counting was used to discover the cellular basis for the decreasing stem cell fractions. For human platelet lysate, it was attributable to lower rates of self-renewing symmetric stem cell divisions. For human sera, both low rates of symmetric division and high rates of stem cell death were responsible. These results demonstrate the power of the KSC counting method to provide previously inaccessible information for improving future tissue stem cell biomanufacturing.

## 1. Introduction

Though there are many products in the regenerative medicine marketplace advertised for promoting the expansion of tissue stem cells for research and stem cell therapeutics, there have been few tools available for evaluating their specific effects on the stem cells in their preparation. Tissue stem cell expansion products include both cell culture medium supplements and cell bioreactor systems.

Nearly all assays in current use for certifying the effectiveness of these products provide information about the effects on committed progenitor cells and differentiated cells as well as stem cells, because these non-stem cell types are often major cellular components in stem-cell-containing preparations, which are typical heterogenous cell populations [1,2,3]. This shortcoming applies to flow cytometry, colony-forming unit assays, and enzyme biomarker assays, because none of these methods quantify tissue stem cells specifically [4,5,6].

The one previously available method with sufficient specificity to estimate the stem-cell-specific fraction of tissue cell preparations is the SCID mouse repopulating cell (SRC) assay. The SRC assay requires large numbers of mice, takes 16 weeks to complete, requires technically demanding flow cytometry, and has poor quantitative resolution [7,8]. Therefore, although now available for two decades, the SRC assay has proven impractical for routine use in quantifying tissue stem cells, and as such remains impractical for evaluating specific effects of growth factors on them in biomanufacturing processes. Moreover, the SRC assay is only applicable for hematopoietic stem cells.

A new method was recently reported for accurately quantifying the specific fraction of stem cells in diverse human tissue cell preparations [9,10]. The new method, called kinetic stem cell (KSC) counting, uses computational simulation to define the number of stem cells in complex heterogeneous tissue cell samples based on the rates at which the stem cells produce other cells during long periods of serial culture [9].

In addition to the initial stem-cell-specific fraction of an evaluated sample, KSC counting can be used to measure how the stem cell fraction of a sample changes during serial culture [9,10]. Such data provide an ideal quantitative basis for assessing the relative effectiveness of different commercial products marketed for their stem cell expansion capabilities. This report provides an instructive example of how KSC counting can be used to define and compare the potency of four different widely used cell culture medium supplements for expanding stem cells in adipose-derived cell preparations containing mesenchymal stem cells (AdMSC).

The reported KSC counting analysis does more than provide unique information that can be used to delineate the best products for expanding tissue stem cells. It also yields actionable insights to the cell kinetics mechanisms responsible for differences in the tissue stem cell expansion potency of different products. These new capabilities should greatly accelerate progress in tissue stem cell biomanufacturing and the regenerative medicine industry that it supplies.

## 2. Materials and Methods

### 2.1. Cells

Four cryovials of StemPro^™^ Human Adipose-derived Stem Cells (#R7788155) from a single donor were purchased from Thermo Fisher Scientific (Cambridge, MA, USA). The cells were thawed rapidly at 37 °C, pooled, and divided equally to initiate the described serial cultures with different growth factor supplements.

### 2.2. Cell Culture Medium and Growth Factor Supplements

The basal culture medium for all serial cultures was MesenPro RS^™^ Basal Medium (#1276012) purchased from Thermo Fisher Scientific and supplemented with 2 mM L-glutamine and 1% penicillin-streptomycin. The following growth factor sources were supplemented as indicated below for evaluation:MesenPro RS^™^ Growth Supplement purchased from Thermo Fisher Scientific and supplemented as instructed by the supplier (“TFS”).10% *v*/*v* fetal bovine serum purchased from Gibco-Thermo Fisher Scientific (#12662-011; “FBS”).10% *v*/*v* pooled human sera purchased from Sigma-Aldrich (St. Louis, MO, USA; #H3667; “HS1”).10% *v*/*v* pooled human sera purchased from Innovation Research (Upper Marlboro, MD, USA; #IPLA-SER; “HS2”).5% *v*/*v* PLTGOLD^®^ clinical grade human platelet lysate (hPL) purchased from Biological Industries USA (Captivate Bio, Watertown, MA, USA; #PLTGOLDXXXGMP; “hPL1”).10% *v*/*v* clinical grade hPL purchased from Cook Regentec (Indianapolis, IN, USA; #G35220/G35221; “hPL2”).

### 2.3. Serial Culture

Triplicate serial cultures were passaged and counted for each of the six different growth factor supplement sources. All serial cultures were initiated by seeding 90,000 viable cells into the wells of six well plates containing 5 mLs of TFS-supplemented culture medium in a humidified 37 °C incubator with a 5% CO_2_ atmosphere. After allowing 5 h for cells to attach, triplicate sets of wells were replaced with culture medium containing each of the six respective different growth factor supplements. Thereafter, cultures were passaged as described previously for KSC counting [10] using TrypLE (Thermo Fisher Scientific; #14200075) for detachment and transferring 1/5 of the detached cells every 72 h of culture (i.e., with a 1:5 split basis). Live cells, based on trypan blue dye exclusion, and dead cells, based on trypan blue dye uptake, were counted using a hemocytometer slide.

### 2.4. KSC Counting Software Analyses

Asymmetrex’s TORTOISE Test^®^ KSC counting software [10] was used to determine the tissue stem cell-specific fraction of serially cultured cells with passage and other cell-type-specific cell kinetics factors as previously described [9,10]. All KSC counting analyses were based on 10 independent computer simulations of experimental cumulative population doubling (CPD) data from the described triplicate serial cultures. Each optimal simulation resulted from 1000 searches by the TORTOISE Test^®^ software for sets of initial stem cells, committed progenitor cells, and terminally arrested cell fractions and cell kinetics factor values that were used to compute the observed experimental data based on a tissue stem cell kinetics cell turnover model [9].

The initial cell kinetics factors determined by KSC counting have been described in detail previously [9,10]. In brief, they are:Stem cell-specific fraction (SCF).Stem cell symmetric self-renewal rate (RS, the rate at which stem cells divide to produce two stem cells).Stem cell-specific death rate (RDS).Committed progenitor cell (CPC)-specific death rate (RDT).Cell cycle time for asymmetrically self-renewing stem cells (GTA, the cell cycle time of stem cells when they divide asymmetrically, producing a stem cell and a first-generation CPC).Cell cycle time for symmetrically self-renewing stem cells (GTS, the cell cycle time of stem cells when they divide symmetrically, producing two stem cells).Cell cycle time for transiently amplifying CPCs (GTT).Turnover division number (TDN, the number of divisions by CPCs before they divide to produce a lineage generation of terminally-arrested cells).

Initial cell kinetics factors determined by the TORTOISE Test^®^ software were input into the RABBIT Count^®^ software [10] to derive Rabbit CPD algorithms as described previously [9,10]. The algorithms were derived with 10 independent computations of the experimental CPD data that gave simulations with fractional root-mean-squared error (fRMSE) values that were ≤0.10. fRMSE is the RMSE of a CPD data simulation compared with the experimental CPD data divided by the maximum value of the experimental CPD data.

Statistical and graphical analyses of KSC counting software outputs were performed with 2020 GraphPad Prism 9 for macOS software, version 9.0.0 (GraphPad Software, Boston, MA, USA).

## 3. Results

### 3.1. Comparative Analysis of Differences in the Cell Proliferation Kinetics of AdMSC-Containing Cells Serially Cultured with Different Growth Factor Supplements

There are a number of proprietary and non-proprietary growth factor supplements currently marketed for use in expanding the number of tissue stem cells in biomanufacturing processes. This report details a study designed to evaluate four commercial supplements widely used for expanding MSCs. Although presumed to have effects on the viability or proliferation of stem cells in MSC-containing cell cultures, such specific effects had not been confirmed previously [11,12,13,14].

The evaluated supplements included one commercial proprietary supplement and three commercial generic supplements that are produced by several different companies. The proprietary supplement, TFS (See Section 2.2), was purchased from the supplier Thermo Fisher Scientific. Throughout the rest of this report, TFS will be referred to as the “Control” in comparison to the non-proprietary supplements, which included one commercial source of fetal bovine serum (FBS), two commercial sources of pooled human sera (HS1 and HS2), and two commercial sources of human platelet lysate (hPL1 and hPL2). Each growth factor source was supplemented as instructed by its supplier.

The recently described KSC counting TORTOISE Test^®^ computational simulation software [9,10] is able to discern independent cell kinetics factors that define the viability and proliferation of the stem cells, transiently dividing committed progenitor cells, and terminally-arrested cells that compose heterogeneous tissue cell samples such as preparations containing commercial AdMSC. The required input for the software is cumulative population doubling (CPD) data and dead cell fraction (DCF) data [9,10]. These data were obtained from triplicate serial cultures developed with the same pool of AdMSC-containing tissue cells cultured in media supplemented with the respective growth factor products. All the cultures were passaged and counted in parallel at each passage. The cultures were passaged with a 1:5 split basis every 3 days.

Figure 1 shows the mean CPD data for each of the six supplement conditions. All of the serial cultures eventually achieved a terminal cell proliferation arrest, which is also a requirement for the KSC counting software analysis [9,10]. Differences in the mean CPD kinetics among the cultures for different supplements were apparent.

The Control, FBS, and hPL2 supplements exhibited similarly high rates and extents of cell proliferation (Compare Figure 1A,B,F). In contrast, human sera supplements, HS1 and HS2, supported significantly less cell proliferation (Compare Figure 1C,D). Cultures supplemented with hPL1 had an intermediate level of cell proliferation (Figure 1E).

The DCF data were determined in parallel at each passage. Two inputs from the DCF data are utilized by the KSC counting software to account for the total number of dead cells produced, which are predominantly terminally-arrested cells present in the cultures, and the variability in their production [9]. These are, respectively, the overall average DCF (Figure 2A–F, Lines) and the mean coefficient of variation of the DCF means determined at each passage (Figure 2A–F, mCOV). As shown in Figure 2, a characteristic feature of all the culture supplements is increasing DCF and DCF variability at later serial passages. The average DCFs for the Control-, FBS-, and human-sera-supplemented cultures were less than 0.1 (Figure 2A–D), whereas the average DCF values for the human platelet lysate supplements were somewhat higher (Figure 2E,F).

### 3.2. KSC Counting Analysis of the Effects of Different Growth Factor Supplements on the Initial Cell Kinetics Factors in the Tissue Stem Cell Kinetics Turnover Unit Model

The CPD data and DCF data were input into the KSC counting TORTOISE Test^®^ software to search for the sets of initiating cell kinetics factors required for simulation of experimental CPD data by using an underlying stem cell kinetics turnover unit model to compute CPD simulation data [9]. Examples of the simulations achieved for the CPD data of each supplement condition are shown in Figure 3.

**Figure 3 life-13-00614-f003:**
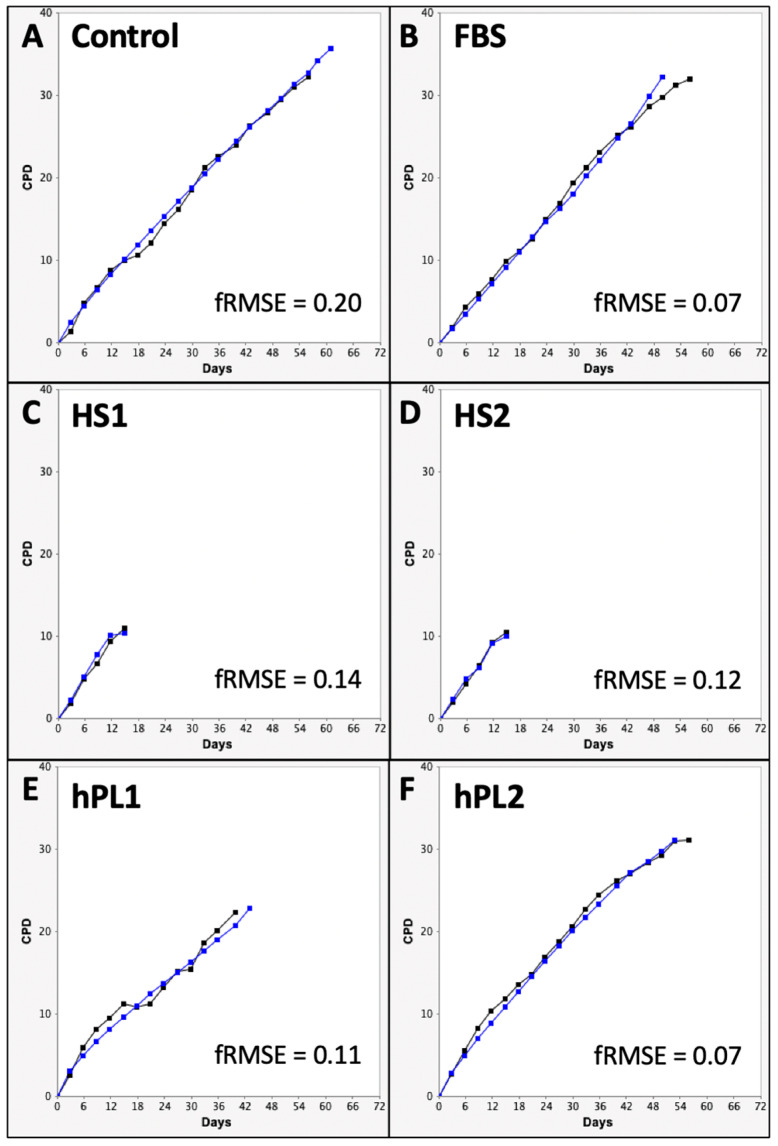
Examples of KSC counting software simulations for experimental mean cumulative population doubling (CPD) data from human AdMSC-containing serial cultures supplemented with the commercial growth factor sources. (**A**) Control; (**B**) FBS; (**C**) HS1; (**D**) HS2; (**E**) hPL1; (**F**) hPL2. Black lines, experimental mean CPD data (from Figure 1). Blue lines, simulated mean CPD data. y-axes, mean CPD; x-axes, days of serial culture. fRMSE values are given as indicators of the quality of the depicted simulations’ approximation of the experimental mean CPD data. fRMSE = root-mean-squared error/maximum mean CPD value. (fRMSE is a different metric than the simulation quality score (SQS) in Figure 4).

With the exception of the hPL1 analysis, all the KSC counting analyses achieved an ideal simulation quality score (SQS) <0.5 (Figure 4, SQS). However, the hPL1 SQS value of 0.65 was still sufficiently below the cut off of 1.0 (on a scale of 100) required for obtaining a confident determination of the set of initial cell kinetics factors [9]. The SQS is a quantitative statistical measurement of how well the experimental CPD data can be described by the stem cell kinetics turnover unit model diagrammed in Figure 4.

**Figure 4 life-13-00614-f004:**
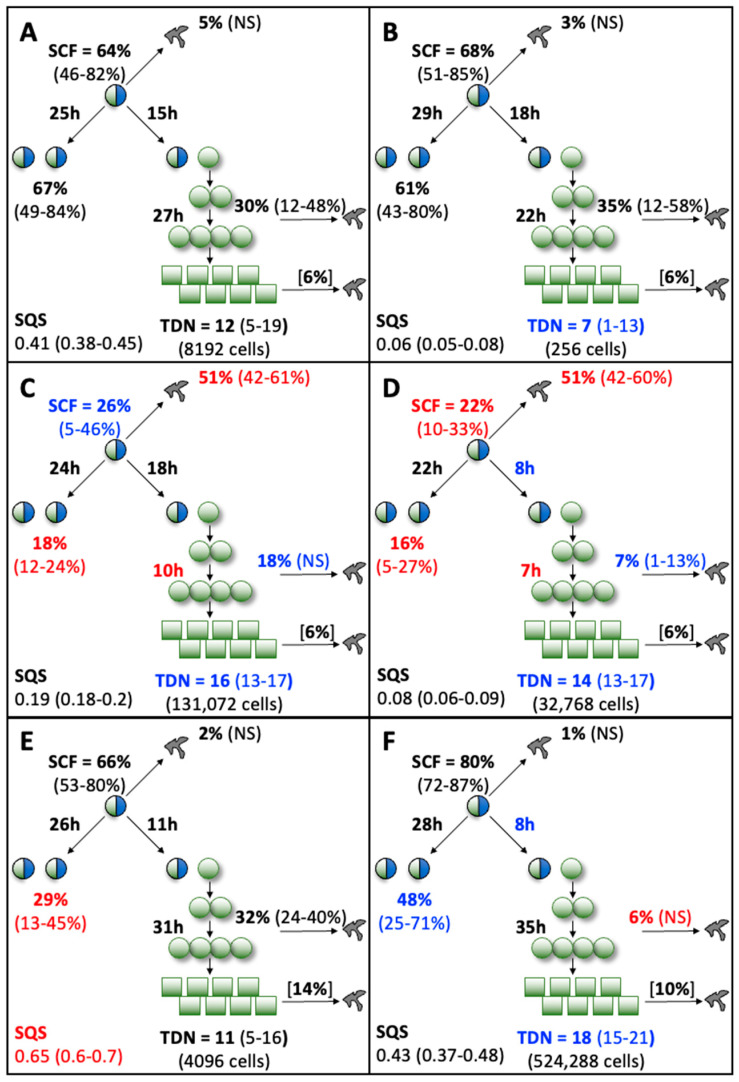
Initial cell kinetics factors defined by KSC counting for the tissue stem cell kinetics turnover unit model. Each diagram depicts the initial cell kinetics factors determined for the evaluated human AdMSC-containing serial cultures supplemented with commercial growth factor sources. Symmetric stem cell divisions (**left**) and asymmetric stem cell divisions (**right**) can be distinguished by their cell products. Bivalent circles, stem cells; uniform circles, transiently-dividing committed progenitor cells; squares, terminally-arrested cells, amorphous shapes, dead cells. %, percent of cells of the labeled type at the start of culture. (#-#%), 95% confidence intervals. NS, not significantly different than 0.0. SCF, stem cell-specific fraction. h, indicated cell cycle times in hours. TDN, turnover division number. (# cells), number of cells in a complete turnover unit. SQS, simulation quality score. (**A**) Control; (**B**) FBS; (**C**) HS1; (**D**) HS2; (**E**) hPL1; (**F**) hPL2. Red text, values significantly different than the respective Control value. Blue text, values noticeably different than the respective Control value, but not statistically significant.

A powerful capability of the KSC counting method is its ability to discover an important set of initial cell kinetics factors that determine the production rates and loss rates during serial culture of the three principal subtypes of cells present in all primary tissue cell preparations [9]. This set of factors includes confident estimates of the respective fractions, division rates, and death rates of the stem cells, transiently-dividing committed progenitor cells, and terminally-arrested cells present at the initiation of a primary tissue cell culture (See Figure 4).

The initiating factor set also includes estimates of two other important cell kinetics factors that govern the character of CPD data. These are the frequency at which stem cells divide symmetrically producing two stem cells and the number of divisions undergone by committed progenitor cells before terminally arrested cells are produced. The second factor is called the turnover division number (Figure 4, TDN). Together, the initiating cell kinetics factors define the respective tissue stem cell kinetics turnover unit models diagramed in Figure 4, which are the biological foundation for the KSC counting method [9].

For simplicity in the evaluation of the observed differences in initial cell kinetics factors associated with different supplements, the proprietary supplement was designated as the Control basis for comparison to the non-proprietary supplements (See red text and blue text in Figure 4). Since all cultures were started with the same pool of AdMSC-containing cells, initial stem-cell-specific fraction (Figure 4, SCF) determinations should be the same for all conditions. This was the case for the Control, FBS, and hPL supplements, whose SCFs, which ranged from 64% to 80%, were not significantly different.

In contrast, cultures supplemented with human sera had SCFs that were remarkably lower at 26% and 22%, respectively, with the lower value of HS2 being a statistically significant difference compared to the Control SCF. The cause of the lower SCF values can be attributed to significantly higher rates of stem cell death detected in cultures supplemented with human sera. Whereas the rates of stem death for other supplements ranged for 1–5% and were not significantly different than a 0% rate, both sources of human sera had estimated mean stem cell death fractions of 51% (Figure 4C,D).

Cultures supplemented with the two sources of human sera showed other differences in common from the Control supplemented cultures. Noticeably, they had statistically significant lower rates of symmetric stem cell division. The respective symmetric division rates of the HS1 and HS2 supplemented stem cells were 18% and 16% compared to the Control supplemented stem cells’ rate of 67%. Though not achieving statistical significance, the human sera supplemented cultures also had lower rates of progenitor cell death and larger TDNs. The respective progenitor cell death fraction and TDN for the Control supplemented cells were 30% and 12. The respective values for the HS1 and HS2 supplemented progenitor cells were 18%, 16 and 7%, 14 (Figure 4C,D). In addition, supplement HS2 had a noticeably shorter cell cycle time for asymmetric stem cell divisions, though the difference (8 h vs. 15 h) was not statistically significant (Figure 4D).

The only difference noted between Control supplemented cells and FBS supplemented cells was a smaller TDN with FBS (7 vs. 12), which did not show statistical significance (Figure 4B). Cells supplemented with hPL1 also showed only one difference. Their rate of stem cell symmetric division was significantly less, 29% vs. 67% for Control supplemented cells (Figure 4E). Though not statistically significant, hPL2 supplemented cells also had a noticeable lower rate of stem cell symmetric division (48%; Figure 4F) compared to Control supplement cells. In addition, hPL2 supplemented cells (Figure 4F) shared three of the differences noted for human sera supplemented cells. These included a statistically significant lower rate of progenitor cell death (6% or less vs. a Control value of 30%); and though not statistically significant, a shorter cell cycle time for asymmetric stem cell divisions (8 h vs. 15 h) and a larger TDN (18 vs. 12).

### 3.3. KSC Counting Analysis of Cell Subtype-Specific Kinetics Profiles Associated with Different Growth Factor Supplements during Serial Culture

The computational features of the KSC counting TORTOISE Test^®^ software provide the ability to evaluate individual changes in the fractions of stem cells, transiently dividing committed progenitor cells, and terminally arrested cells as specified by the stem cell kinetics turnover unit model [9,10]. Shown in Figure 5, these analyses make readily apparent remarkable differences and similarities in the supplements’ effects on the serial culture kinetics of each cell subtype. The Control, FBS, and hPL supplements were similar in supporting high fractions of committed progenitor cells (Figure 5A,B,E,F, red traces) with late development of terminally-arrested cells (Figure 5A,B,E,F, green traces). In contrast, both human sera sources had high fractions of terminally-arrested cells in their cultures for the start (Compare Figure 5C,D, green traces) and produced fewer committed progenitor cells (Figure 5C,D, red traces). All the supplements showed the characteristic decline in the SCF with serial passage (Figure 5, blue traces) [9,10,15,16,17,18,19,20]. However, the decline in SCF was much less pronounced for Control supplemented and FBS supplemented cultures (Figure 5A,B, blue traces).

### 3.4. Comparion of the KSC Counting-Derived Rapid Stem Cell Counting Algorithms Associated with Different Cell Culture Growth Factor Supplements

The KSC counting RABBIT Count^®^ software was used to derive Rabbit CPD algorithms for evaluation [10]. These recently described mathematical algorithms provide rapid determinations of the SCF of cultures of similar primary tissue cell preparations when they are cultured under the same conditions as those used to derive the algorithms. Rabbit CPD algorithms compute the SCF at any time during serial culture based on a culture’s number of CPDs. Differences in the mathematical form of Rabbit CPD algorithms derived for the same tissue cell source cultured under different conditions reflect differences in the underpinning tissue cell kinetics.

Figure 6 shows the graphical outputs of the Rabbit CPD algorithms derived for AdMSCs in cultures supplemented with the different growth factor products. Supplements of the same class had CPD algorithms of similar character that were distinct from the algorithms of other supplement classes. The CPD algorithms for human sera were similar to each other (Compare Figure 6C,D), but differed from human platelet lysate algorithms, which in turn were more similar to each other (Compare Figure 6E,F). These relationships differences were quantified by their respective stem cell fraction half-lives (SCFHL; Figure 6). The SCFHL is the number of CPDs required for the SCF of a serial culture to decline by 50%. The SCFHLs of human sera supplemented cultures, 0.65 CPD and 0.70 CPD (Figure 6C,D), were significantly shorter that the SCFHLs of the human platelet lysate supplemented cultures, which were 1.63 CPD and 1.39 CPD (Figure 6E,F), respectively.

Paralleling their similar more stable SCF with passage (See Figure 5A,B), the CPD algorithms of Control supplemented and FBS supplemented AdMSCs showed a high degree of SCF maintenance with increasing number of CPDs (Figure 6A,B). The estimated SCFHL for FBS supplemented AdMSCs of 4.82 CPD was significantly longer than the SCFHLs of human sera supplemented and human platelet lysate supplemented AdMSCs. The character of the CPD algorithm for Control supplemented AdMSCs was highly similar to the form of the FBS CPD algorithm. However, its estimated SCFHL was deceptively shorter (Figure 6A, 1.69 CPD), because it applied only to an initial decline phase before the SCF of Control supplemented AdMSCs achieved a stable plateau with increasing number of CPDs.

## 4. Discussion

The reported study was undertaken to address two entwined major challenges that continue to limit the rate of progress in stem cell science and stem cell medicine, with the goal of accelerating progress in the biomanufacturing of therapeutic tissue stem cells. The challenges are (1) the specific quantification of the fraction and dosage of tissue stem cells and (2) achieving their expansion in culture without loss of stemness potency. These are entwined challenges because, without a means to quantify and monitor the number of potent stem cells specifically, it has been extremely difficult to faithfully engineer their expansion [16,17,18,19,20].

These two long-standing challenges reflect the inherent cell heterogeneity of vertebrate animal tissues, including those of humans. The significant cell heterogeneity of both primary and processed human tissue cell preparations is well appreciated, and it has been considered as an important issue for attention in human cell therapy development [1,2,3]. However, its importance as a key factor making the quantification and expansion of tissue stem cells so challenging is not widely recognized. This potential paradox may be due to a deficit in general academic and industry understanding that many of the in vivo cell kinetics properties of tissue cell populations continue in cell culture [15,16,17,18,19,20].

The KSC counting method used for the presented study was developed based on this important principle that, although many cell differentiation properties are lost during cell culture, the fundamental tissue cell renewal kinetics are maintained [9,10]. The in vitro stem cell-based turnover unit model (See Figure 4) is based on how tissue stem cells in vivo primarily divide asymmetrically to renew themselves while simultaneously producing non-stem committed progenitor cells [21,22]. Although in cell cultures the committed progenitor cells produced lose many of their in vivo differentiation phenotypes, they retain the transient division property and produce a final generation of cells that undergo a terminal division arrest [15]. This general in vitro cell renewal paradigm can account for the cell kinetics features that constitute the well-known Hayflick limit of primary human tissue cell cultures [15,23].

Based on this underlying cellular model, the KSC-counting TORTOISE Test^®^ software is not only able to define the initial stem cell fraction of a heterogenous tissue cell preparation, but it is also able to compute the fractions and kinetics of all three cell kinetics subtypes during serial cell culture. Previously, the specificity, accuracy, and reliability of KSC counting has been validated by several orthogonal tests [9,10], including most recently by comparison to the SCID mouse repopulating cell assay [10], which is the only other method available for determining the specific fraction of a human tissue stem cell type [7,8].

Although the presently reported study focused on adipose-derived MSCs, KSC counting has general application to stem cells in any type of renewing vertebrate tissues [9,10]. Previous studies have quantified tissue stem cells from other human tissues, including hematopoietic stem cells [9,10], liver stem cells [9], and MSCs from other tissue sources [9]. In unpublished studies, tissue stem cells have been quantified in cell preparations from human oral tissues (gingival, mucosal, and dental bone), umbilical cord tissue, and cornea. To date, animal tissues have not been investigated. However, because of the universal asymmetric self-renewal by stem cells in vertebrate tissues [22], KSC counting has been considered for applications in animal research and veterinary medicine as well [6]. Similarly, for cancer stem cells that maintain asymmetric self-renewal kinetics, KSC counting may be effective for their quantification. In the case of pluripotent stem cells, which lack asymmetric self-renewal kinetics, KSC counting does not have application, but it is also not needed. Unlike the challenge of counting unipotent and multipotent tissue stem cells, there are many highly specific molecular biomarkers available for identifying and quantifying pluripotent stem cells, including both embryonic stem cells and induced pluripotent stem cells.

The focus of the study presented in the present report was the evaluation of the specific effects of commercial growth factor supplements widely used with the purpose of expanding stem cells in tissue cell preparations containing human MSCs. MSC-containing preparations, including those derived from adipose tissue, exhibit a common decline in stem cell-attributed properties with increasing passages in culture [24,25,26,27]. The initial report of KSC counting provided the first indication that this decline in stem cell function was associated with a decline in the SCF of passaged MSC-containing preparations derived from human bone marrow [9]. The present report extends this fundamental advance in understanding to explain the loss of MSC phenotypes with the passage of human-AdMSC-containing cell preparations.

The study described in this new report adds to the evidence supporting the concept that much of the loss of tissue stem cell function during primary cell culture is not due to phenotypic changes in the stem cells per se. Instead, it can largely be accounted for by decreases in the stem cell fraction as a result of their asymmetric production of differentiated cells, which, in the case of dividing committed progenitor cells, are a major cause of their loss by cell dilution with increasing passages.

The presented results show that KSC counting not only provides a means to quantify and monitor changes in the SCF of AdMSCs during expansion processes, but it also is a powerful tool for revealing the cell kinetics basis for the action of growth factor supplements used for MSC biomanufacturing. This unique capability is most evident in the analyses of the Control-, FBS-, and hPL2-supplemented cultures. In the commonly held view, based on their similar degrees of total cell proliferation and similar overall low cell death rates, all three of these supplements would have been considered to have similar effects on AdMSC expansion. However, on closer inspection, it is now noteworthy that the hPL2 mean CPD data are slightly hyperbolic. (Compare the mathematical form of Figure 1F data to Figure 1A,B.) The higher-resolution KSC counting TORTOISE Test^®^ software analysis revealed that, in fact, hPL2 is neither expanding nor preserving AdMSCs. The hPL2 supplement fails to support the high rates of AdMSC symmetric division, supported by the Control and FBS supplements, which is essential for AdMSC maintenance and expansion. Instead, the high rate of cell proliferation by hPL2 supplemented cultures is attributable to faster asymmetric cycling by AdMSCs, with increased production of longer-lived committed progenitors that also divide more extensively before undergoing terminal division arrest. (Compare Figure 4F to Figure 4A,B.) These new insights align well with earlier reports of human platelet lysate inducing greater levels of differentiating cell phenotypes in cultures containing MSC compared with FBS [28,29]. Other reported differences between the effects of human platelet lysate and FBS [30] may be related to differences in the cell subtype distributions defined by KSC counting.

In a similar fashion, the KSC counting analysis gives actionable insights into the lesser potency of human sera in the presented studies [31]. The poor overall proliferation of these cultures is attributable to significantly lower rates of AdMSC symmetric division and significantly higher rates for AdMSC death, which was not evident from the overall rate of cell death in the cultures (See Figure 4C,D). Knowledge of these cell-type-specific cell kinetics mechanisms provides the opportunity for devising specifically targeted interventions to increase symmetric stem cell divisions and stem cell viability. Furthermore, KSC counting provides the means to evaluate the effects of such evaluated interventions on each individual cell subtype present in complex tissue cell cultures.

The recently defined SCFHL has the potential to become an effective basis for comparing the effectiveness of different cell expansion conditions to stabilize or even increase the SCF of biomanufacturing cultures. This new state parameter quantifies the integration of the effects of the several cell kinetics factors that determine the stability of the SCF during serial culture. The SCF of a culture depends on the death rates and production rates of the stem cells in the culture, of their progeny committed progenitor cells, and of their descendant terminally-arrested cells. A high value of the SCFHL is required to achieve net production of tissue stem cells. Higher values of the SCFHL can be obtained by the following main strategies: maintaining high stem cell viability, inducing high rates of symmetric stem cell division, and reducing the proliferation of committed progenitor cells. The KSC counting method not only informs this innovative basis for more effective engineering of tissue stem cell biomanufacturing, but it also provides the quantitative tool needed for implementing it.

## Figures and Tables

**Figure 1 life-13-00614-f001:**
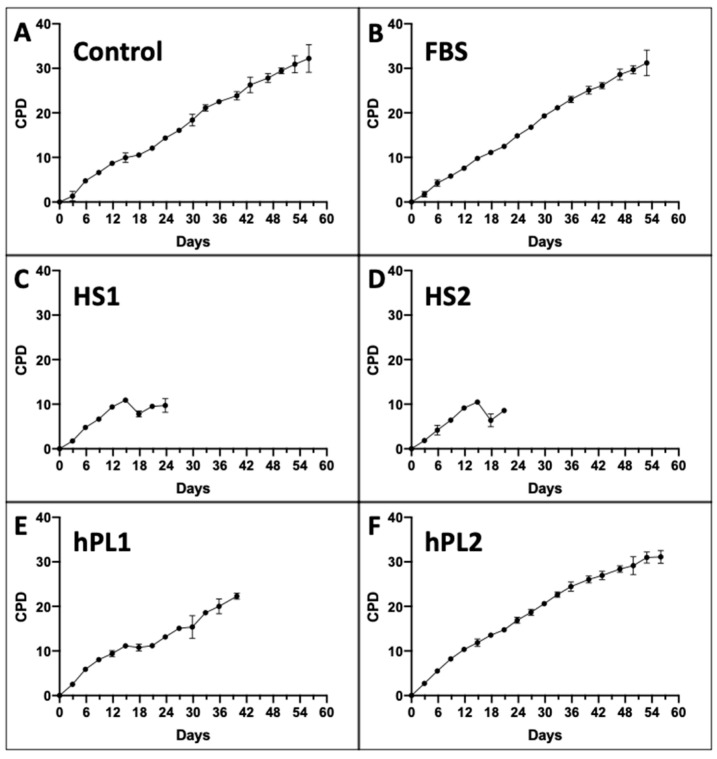
Cumulative population doubling (CPD) data used for input into KSC counting software analyses. Shown are graphs developed with CPD data from triplicate serial cultures of human AdMSC-containing tissue cells supplemented with the different commercial sources of growth factors. Every three days, culture cells were detached, counted, and passaged 1:5. All serial cultures were continued until no increase in cell number was detected. (**C**,**D**) the data for HS1 and HS2 cultures were continued for additional passages beyond their terminal arrest to verify that proliferation did not resume. The data are reported as the mean of the CPD data for the triplicate cultures. (**A**) Control; (**B**) FBS; (**C**) HS1; (**D**) HS2; (**E**) hPL1; (**F**) hPL2. y-axes, mean CPD; x-axes, days of serial culture; error bars, 95% confidence interval about the CPD mean values.

**Figure 2 life-13-00614-f002:**
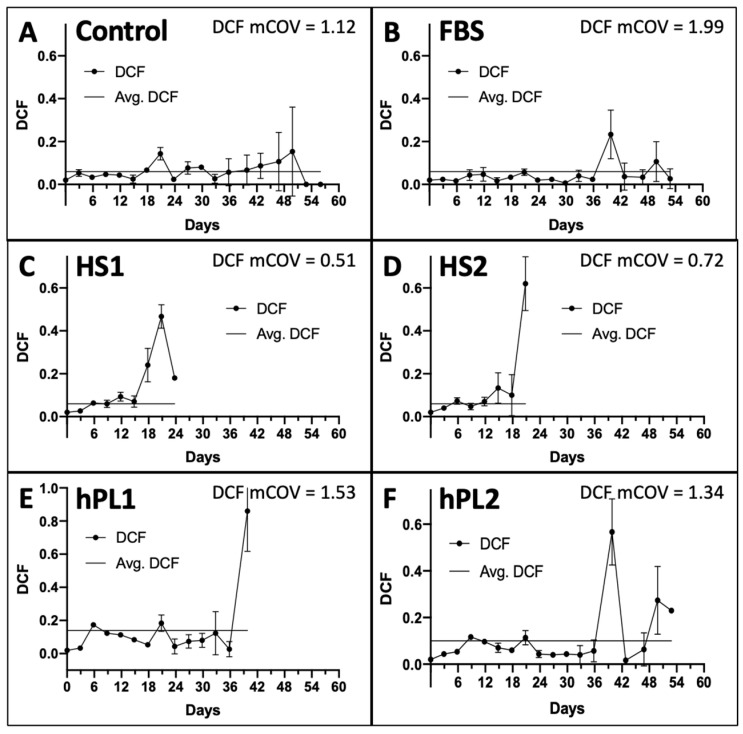
Dead cell fraction (DCF) data used for input into the KSC counting software analyses. Shown are the respective graphs developed with trypan-blue-positive DCF data determined at each passage of the serial cultures described in Figure 1. The data are reported as the mean of the DCF data for the triplicate cultures. (**A**) Control; (**B**) FBS; (**C**) HS1; (**D**) HS2; (**E**) hPL1; (**F**) hPL2. y-axes, mean DCF; x-axes, days of serial culture; error bars, standard deviation. Line, the overall average DCF value. mCOV, the overall mean of the coefficient of variations for the mean DCFs at each passage. (**C**,**D**) the mean DCF and mCOV for HS1 and HS2 cultures only pertain to the first five passages, whose data were used for simulation in KSC counting software analyses (See Figure 3).

**Figure 5 life-13-00614-f005:**
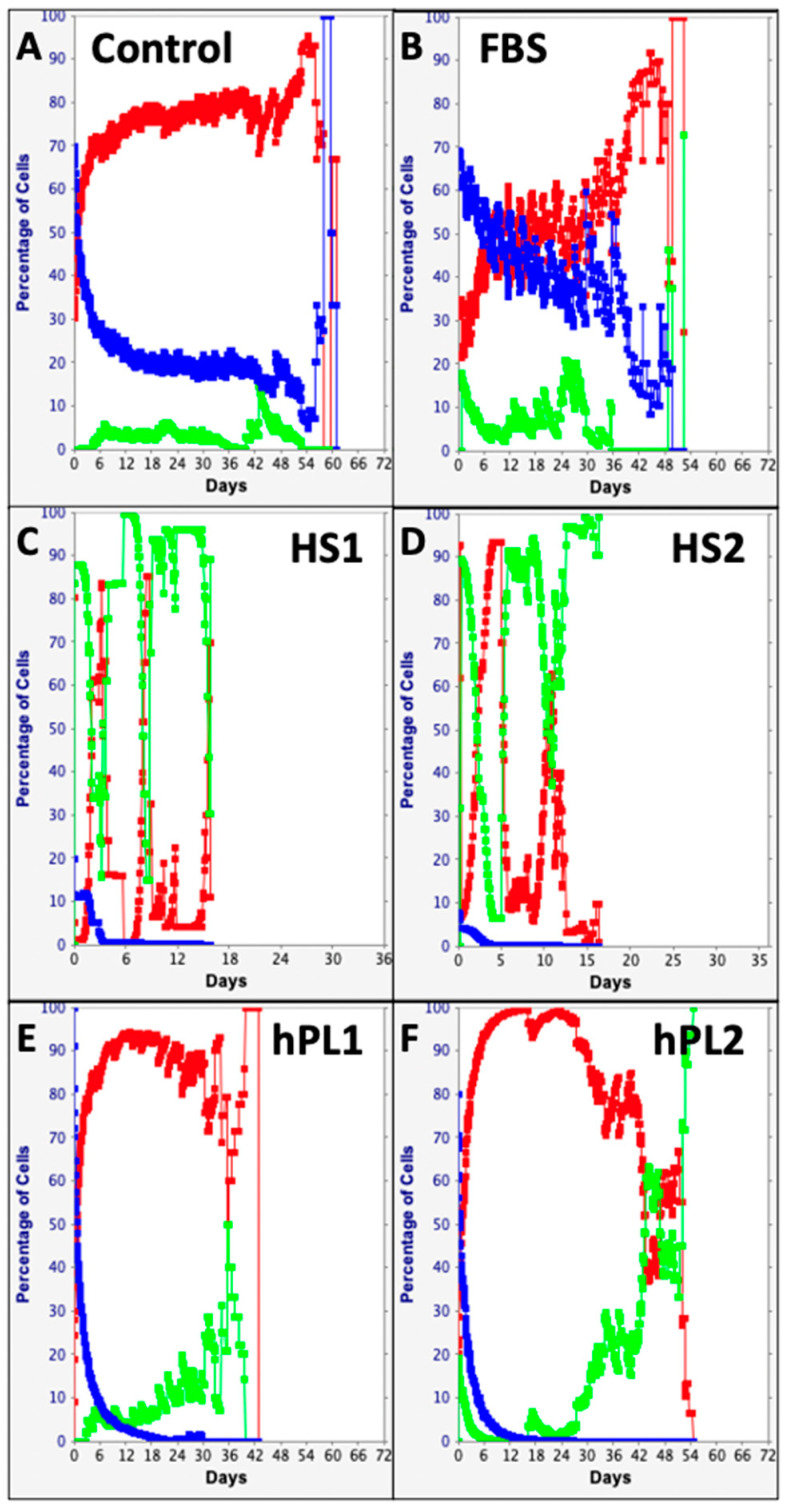
Cell subtype-specific cell kinetics profiles computed with initial cell kinetics factors defined in KSC counting analyses for human AdMSC-containing cultures supplemented with commercial growth factor products. Shown are the calculated individualized cell kinetics data during passaging for stem cells (blue), transiently-dividing committed progenitor cells (red), and terminally-arrested cells (green) in terms of their percent cellularity. The calculations were performed with the respective sets of initial cell kinetics factors in Figure 4; and they correspond to the respective KSC counting mean CPD simulations presented in Figure 3. (**A**) Control; (**B**) FBS; (**C**) HS1; (**D**) HS2; (**E**) hPL1; (**F**) hPL2.

**Figure 6 life-13-00614-f006:**
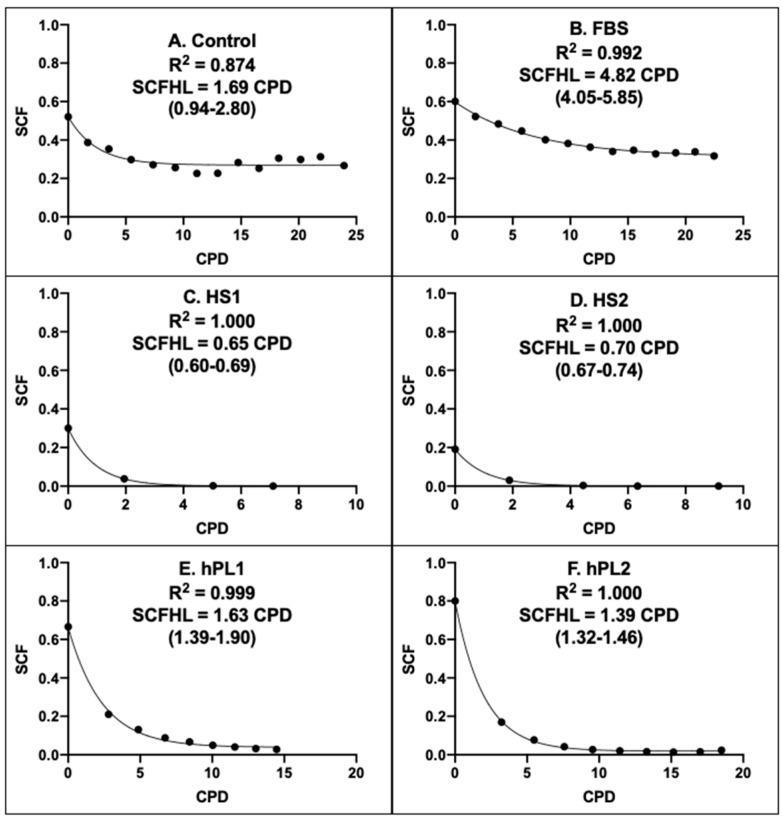
Rabbit CPD algorithms for rapid determination of the stem-cell-specific fraction of AdMSC-containing cell preparations. Each panel provides a graphical representation of the KSC counting-derived CPD algorithm that can be used to rapidly calculate or project the stem cell-specific fraction (SCF) of human AdMSC-counting cell preparations, cultured with the respective commercial growth factor products, based on a culture’s number of cumulative population doublings (CPD). (**A**) Control; (**B**) FBS; (**C**) HS1; (**D**) HS2; (**E**) hPL1; (**F**) hPL2. R^2^, Pearson correlation coefficient; SCFHL, SCF half-life; (#-#), 95% confidence interval. Note: The SCFHL in panel A corresponds only to the initial decline phase that plateaus at CPD = 9.

## Data Availability

All the data for the study are in the figures in the manuscript.

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
