# Peer review of "A Kinetic Stem Cell Counting Analysis of the Specific Effects of Cell Culture Medium Growth Factors on Adipose-Derived Mesenchymal Stem Cells"

_life, 2023, doi:10.3390/life13030614_

Round 1
Reviewer 1 Report
In this study, the authors used the newly reported method, kinetic stem cell (KSC) counting, to assess the effects of commercial culture factors on the growth of fat-derived tissue stem cells. Based on their analysis, this new method may provide more effective ways to improve tissue stem cell biomanufacturing in the future. While this study is interesting, I still have several concerns and my detailed comments are listed as following.
1. The authors should also label the supplement conditions in each figure, though they clarified them in the figure legends. In addition, the authors may consider merging figures A-F to one figure and mark different conditions in color to make it easier for comparison.
2. For Figures 1C-1D, I notice that there are declines in cell proliferation on the 18th day of culturing with HS1 and HS2, then cells restart proliferation. However, based on the results of Figure 2, dead cells do not reach to the maximal levels on the 18th day in both culture conditions. These results seem to contradict the results of Figures 1C-1D. Can the authors explain the reason?
3. Figure 3, in the figure legend, the authors should also specify what black lines represent.
4. Figure 4B, why is the cell number only 256? I have a concern that such a big difference between FBS and other groups may affect the accuracy of analysis. The author should explain it in the manuscript.
5. Figures 5C and 5D, why do all the traces in HS1 and HS2 groups fluctuate intensely over the time?
6. The authors may consider to rewriting the abstract, since the current abstract lacks the brief background and significance of this study.
Author Response
I thank the reviewers for reading the manuscript and offering helpful suggestions and recommendations for its further improvement. I have endeavored to diligently address each of the reviewers’ comments and incorporate their excellent suggestions into the submitted revised manuscript. I hope the described revisions meet with their approval.
Responses for Reviewer 1
- Supplement condition labels were added to all figures. Subfigures A-F for Fig. 1 were not merged, because such a formatting would obscure important individual features of the CPD datasets.
- The reviewer’s observations of the quantitative distinctions of the two sets of data are correct, but there is no discrepancy between them because Fig. 2 data are not total dead cells, but instead dead cell fraction data. Therefore, the reviewer’s comparison is incongruent.
- The Fig. 3 legend does define the black lines: “Black lines, experimental mean CPD data (from Fig. 1).”
- “256” in Fig. 4B, does not refer to the number of cells in the cultures. Instead, it is the number of differentiated cells produced from any one asymmetric stem cell division before all the produced cells undergo a terminal division arrest. There are thousands of these units in the actual cell cultures. The meaning of 256 should be clearer now after the inclusion of more background on the initial cell kinetics factors, which was added to the Materials and Methods section, as recommended by Reviewer 2.
- At this time, we choose to not speculate on the meaning or cause of the noted significant fluctuation between committed progenitor cells and terminally arrested cells, as we do not understand it. These transitions are highly reproducible in cultures that have high rates of stem cell death like the HS1 and HS2 supplemented cells. It may reflect a periodicity in the dying of the stem cells under these conditions, which is something we are investigating now, but have no advance in understanding to report at this time.
- The opening sentence of the abstract is the essential background for the study, the development of KSC counting; and the ending sentence is a clear statement of the study’s importance for improving tissue stem cell biomanufacturing.
Reviewer 2 Report
This study provides new evidence to support the usage of a kinetic stem cell counting system to evaluate the efficacy of various supplements on the stem cell fraction in culture. The aim is to improve methods to assess aspects of stem cell proliferation and population within a given culture sample. The study is well designed, the results are convincing and I only have minor comments. I am sure this method will be helpful for the community.
Minor comments:
1. The methods although described before should be fairly standalone and useful for the generalist reader even if they are doing stem cell culture. Inability to use KSC due to jargon would be problematic for the community. Perhaps in the methods section a brief explanation of the cell type specific kinetics factors would be helpful such that the reader does not have to be going back several publications to understand the data and its relevance.
Figure 4 annotations could be improved. It wasn't quite apparent that the two arrows from SCF meant symmetric and asymmetric division. That can be easily labeled on any one of the panels so that eh figure can be standalone and not confusing, given the number of arrows, numbers and symbols already on it.
Could the authors comment on whether this software can be modified to be applied to other tissue culture and/or in vivo cell populations with a similar sort of population structure? Just a minor discussion point.
Does this software work for all different types of stem cells? For example, would this work for the more problematic embryonic stem cells or spermatocyte stem cells that have vastly different cell division times? This information would be helpful for researchers to evaluate the usage of the software and allow it to be modified for other applications in the future.
Author Response
I thank the reviewers for reading the manuscript and offering helpful suggestions and recommendations for its further improvement. I have endeavored to diligently address each of the reviewers’ comments and incorporate their excellent suggestions into the submitted revised manuscript. I hope the described revisions meet with their approval.
Responses for Reviewer 2
As recommended, a brief overview of the intial cell kinetics factors determined by the KSC counting method has been added to the Materials and Methods section.
Instead of adding additional labels to Fig. 4 diagrams, the following sentence was added to the figure legend: “Symmetric stem cell divisions (left) and asymmetric stem cell divisions (right) can be distinguished by their cell products.” As noted by the reviewer, this specific direction was lacking and needed.
A paragraph was added to the Discussion section to address the reviewer’s remaining suggestions. However, the “in vivo” application query was not considered, because KSC counting’s dependence on serial cell culture makes it inherently an in vitro application.
Reviewer 3 Report
This study shows the use of the KSC counting method for assessing the effect of different cell culture supplements on Ad-MSC culture. The KSC method can be quite useful for characterizing the tissue-specific stem cell growth in culture. But the current study has some limitations as follows,
1. The growth supplements used in this study do not all support the maintenance of Ad-MSC as shown by the CPD data. Human sera clearly does not support the maintenance of MSC.
2. CPD and DCF data clearly show that the TFS and FBS supplements are superior to human sera and platelet lysate supplements and so the use of KSC method must be validated with supplements that have more comparable CPD and DCF.
3. The author needs to describe how the KSC method can account for the differences in characteristics of various types of tissue-specific stem cells in culture.
4. The accuracy of KSC method for predicting SCF must be validated with methods such as flow cytometry.
Author Response
Responses for Reviewer 3 Comments
The author thanks the reviewer for reading the manuscript and offering constructive criticisms. Each of the reviewer’s observations and comments are addressed with the responses below.
- The growth supplements used in this study do not all support the maintenance of Ad-MSC as shown by the CPD data. Human sera clearly does not support the maintenance of MSC.
The reviewer’s observations are the motivation and purpose of the paper. KSC counting can be used to elucidate the underlying stem cell kinetics basis for differences in the effectiveness of commercial growth factor supplements to support expansion of human stem cell-containing preparations.
- CPD and DCF data clearly show that the TFS and FBS supplements are superior to human sera and platelet lysate supplements and so the use of KSC method must be validated with supplements that have more comparable CPD and DCF.
The purpose of the presented study was not validation of the KSC counting method. The method was already validated by multiple orthogonal evaluations reported in two previous reports, references 9 and 10. In addition, in the present study, there are paired similar CPD and DCF datasets (i.e., Control and FBS; HS1 and HS2; hPL1 and hPL2), which yield similar KSC counting results. This concordance implicitly provides further validation of the reproducibility of the method.
- The author needs to describe how the KSC method can account for the differences in characteristics of various types of tissue-specific stem cells in culture.
Only one type of tissue-specific stem cell was evaluated in the present report, adipose-derived mesenchymal stem cells. However, in response to Reviewer 2’s suggestion, the following paragraph was added to the Discussion section. It addresses Reviewer 3’s suggestion as well.
“Although the presently reported study focused on adipose-derived MSCs, KSC counting has general application to stem cells in any type of renewing vertebrate tissues [9,10]. Previous studies have quantified tissue stem cells from other human tissues, including hematopoietic stem cells [9,10], liver stem cells [9], and MSCs from other tissue sources [9]. In unpublished studies, tissue stem cells have been quantified in cell preparations from human oral tissues (gingival, mucosal, and dental bone), umbilical cord tissue, and cornea. To date, animal tissues have not been investigated. However, because of the universal asymmetric self-renewal by stem cells in vertebrate tissues [22], KSC counting has been considered for applications in animal research and veterinary medicine as well [6]. Similarly, for cancer stem cells that maintain asymmetric self-renewal kinetics, KSC counting may be effective for their quantification. In the case of pluripotent stem cells, which lack asymmetric self-renewal kinetics, KSC counting does not have application, but it is also not needed. Unlike the challenge of counting unipotent and multipotent tissue stem cells, there are many highly-specific molecular biomarkers available for identifying and quantifying pluripotent stem cells, including both embryonic stem cells and induced pluripotent stem cells.”
- The accuracy of KSC method for predicting SCF must be validated with methods such as flow cytometry.
As noted and cited in the Introduction, flow cytometry cannot be used to validate KSC counting, because there are no reported flow cytometry biomarkers that detect stem cells independently of committed progenitor cells.
“Nearly all assays in current use for certifying the effectiveness of these products actually provide information about effects on committed progenitor cells and differentiated cells as well as stem cells, because these non-stem cell types are often major cellular components in stem cell-containing preparations, which are typical heterogenous cell populations [1-3]. This shortcoming applies to flow cytometry, colony forming unit assays, and enzyme biomarker assays, because none of these methods quantify tissue stem cells specifically [4-6].”
Therefore, in our previous reports [9,10] we validated the KSC counting method by cell fractionation studies (e.g., CD34 selection [9,10]), time-lapse microscopy [9], in situ cell kinetics cytometry [9], pharmaco-cell kinetics [9], and the SCID mouse repopulating cell assay [10].
Round 2
Reviewer 3 Report
The author has made minor revisions which partially address the concerns.